# Evaluation of Mental Health Factors among People with Systemic Lupus Erythematosus during the SARS-CoV-2 Pandemic

**DOI:** 10.3390/jcm9092872

**Published:** 2020-09-05

**Authors:** Paweł Wańkowicz, Aleksandra Szylińska, Iwona Rotter

**Affiliations:** Department of Medical Rehabilitation and Clinical Physiotherapy, Pomeranian Medical University in Szczecin, Żolnierska 48, 71-210 Szczecin, Poland; aleksandra.szylinska@gmail.com (A.S.); iwrot@wp.pl (I.R.)

**Keywords:** SARS-CoV-2, COVID-19, SLE, GAD-7, PHQ-9, ISI

## Abstract

There are considerable psychological and psychiatric consequences of the pandemic. Researchers have started to take into account the real or perceived sense of social threats that may be expressed, such as anxiety, depression, and sleep disorders. However, analyses on pandemic-related anxiety, depression, and sleep disorders mostly rarely addresses the situation of people with autoimmune diseases. Therefore, the aim of this study was to assess the mental health factors among people with systemic lupus erythematosus by quantifying the severity of anxiety, depression, and sleep disorders during the SARS-CoV-2 pandemic. In total, 723 people took part in the study. The study group consisted of 134 individuals with a systemic lupus erythematosus. The control group consisted of 589 people without systemic lupus erythematosus. The regression adjusted by age, gender, and diagnosis of other chronic diseases showed individuals with systemic lupus erythematosus were at a much higher risk of elevated symptoms of anxiety on the GAD-7 scale (OR = 3.683; *p* < 0.001), depression on the PHQ-9 scale (OR = 4.183; *p* < 0.001), and sleep disorders on the Insomnia Severity Index (ISI) scale (OR = 6.781; *p* < 0.001). Therefore, the mental health of patients with systemic lupus erythematosus in the times of the SARS-CoV-2 pandemic is not only an extremely important medical problem but also a social one and must require special attention.

## 1. Introduction

The entire world is currently trying to cope with the global pandemic of a new coronavirus. This was first observed in the city of Wuhan in China at the end of 2019, when cases of new atypical pneumonia were discovered [1]. The International Committee on Virus Taxonomy called this type of coronavirus SARS-CoV-2 due to its similarity to a virus causing severe acute respiratory syndrome (SARS) [2]. The disease caused by the new virus is commonly referred to as COVID-19 and has a higher mortality rate than that for influenza and, unlike SARS, is far more contagious [3]. To date, SARS-CoV-2 has killed 750,000 people and over 21 million have been infected, causing a global pandemic which has posed a serious challenge to medical care systems worldwide. In Poland, there have been 55,000 confirmed cases and 1800 deaths.

Based on available information and clinical knowledge, the Center for Disease Control and Prevention (CDC) announced that the majority of SARS-CoV-2 infections are asymptomatic or oligosymptomatic except for the elderly and people of all ages with chronic diseases, who may experience severe symptoms of COVID-19 [4]. Depending on the report, chronic diseases occur in up to 50% of patients infected with SARS-CoV-2 and mortality rates in this group are significantly higher than in the average population. During the recent coronaviral epidemics, i.e., SARS and MERS, it was also observed that in the majority of people with chronic diseases, the symptoms were more severe and often led to death due to multi-organ failure [5].

The chronic diseases listed by the CDC as those that can lead to a severe course of COVID-19 include many conditions that can cause immunosuppression, e.g., autoimmune diseases, which are becoming increasingly common although their exact causes are largely unknown. This group of chronic diseases is associated with the dysfunction of the immune system and consists of an undirected reaction against one’s own cells, tissues, and organs [6,7]. This reaction results from a complex interaction between environmental and genetic factors. In the United States alone, autoimmune diseases affect more than 25 million people and their incidence is constantly increasing. Despite global progress in the diagnosis of these diseases, it is still difficult to identify them at the preclinical stage.

There are also considerable psychological and psychiatric consequences of the pandemic, and, in general, other adverse epidemiological conditions created by the industrialized world. Researchers have started to take into account the real or perceived sense of social threats, fear, and uncertainty that may be expressed as anxiety, depression, and sleep disorders [8,9,10,11,12]. However, analyses on pandemic-related anxiety mostly concern health care workers, rarely addressing the situation of other groups, e.g., people with systemic lupus erythematosus (SLE) [13]. The current outbreak of SARS-CoV-2 infection has led to global changes in many dimensions of daily lives. Ubiquitous information about the number of deaths, new diseases, diseases predisposing one to a severe and unfavorable course of SARS-CoV-2 infection, lack of targeted treatment, social isolation, change of existing habits, long-term quarantine, and a limited or total lack of access to goods or medical support causes a huge mental burden by generating emotional stress, elevated anxiety, and sleep disturbances. Sleep disturbances, depression, and anxiety are commonly reported in SLE patients. However, their prevalence varies from study to study [6,14,15]. Therefore, the aim of this study was to assess the mental health factors among people with SLE by quantifying the severity of anxiety, depression, and sleep disorders during the SARS-CoV-2 pandemic.

## 2. Materials and Methods

This cross-sectional study was conducted in the Western Pomerania region in Poland from 3 May 2020 to 17 May 2020. The study area was showing an incidence at 6/100.00 and prevalence at 31/100.00 during this period. This study included 6 hospitals with clinics or wards that diagnosed or hospitalized COVID-19 patients. In total, 723 people took part in the study. The study group consisted of 134 individuals with SLE. The control group consisted of 589 people without SLE. Each of the survey participants gave their informed consent by taking part in the survey. The participants could stop the survey at any time. The survey was anonymous and ensured the full confidentiality of information.

In this study, we focused on the symptoms of anxiety, depression, and sleep disorders in all participants using the 7-item Generalized Anxiety Disorder scale (GAD-7; range 0–21; no anxiety ≤ 4, anxiety > 4) [16] to assess the severity of anxiety, the 9-item Patient Health Questionnaire (PHQ-9; range 0–27; no depression ≤ 4, depression >4) [17,18,19,20,21,22] to assess the severity of depression symptoms, and the 7-item Insomnia Severity Index (ISI; range 0–28; no insomnia ≤ 8, insomnia > 8) to assess the severity of sleep disorder symptoms [23,24,25,26]. Cut-off points for anxiety, depression, and insomnia were established in accordance with the literature. Participants with scores below the cut-off points were characterized as showing no symptoms, while those who obtained scores above the cut-off points were characterized as showing symptoms.

Each participant gave basic demographic data, including their gender (male or female) and age. Data on coexisting diseases such as diabetes mellitus, hypertension, heart failure, coronary heart disease, chronic obstructive pulmonary disease, and dyslipidemia were also collected from each participant. Each participant was also asked about tobacco smoking.

The Pomeranian Medical University Ethics Committee approved the study protocol (KB-0012/26/04/2020/Z) which conformed to the ethical guidelines of the Declaration of Helsinki.

### Statistical Analysis

A licensed Statistica 13.0 program (StatSoft, Tulsa, OK, USA) was used for statistical analysis. The assessment of normal distribution was performed using the Shapiro–Wilk test. The analysis of quantitative data was performed using the Mann–Whitney U test. For the analysis of qualitative data, the *X*^2^ test was used; if the subgroup size was small, the Yates correction was applied. The evaluation of the relationship between the analyzed parameters was performed using univariable logistic regression model analysis and was adjusted for potentially distorting data (age, gender, diagnosed hypertension, diabetes mellitus, dyslipidemia, and cigarette smoking). Statistical significance was set at a *p* ≤ 0.05.

## 3. Results

### 3.1. Comparison of Coexisting Diseases and Basic Demographic Data between the Group of People with Systemic Lupus Erythematosus and People without a Diagnosis of Systemic Lupus Erythematosus

Individuals without a diagnosis of systemic lupus erythematosus significantly more often smoked cigarettes and were more likely to suffer from hypertension (*p* < 0.001, *p* = 0.003, respectively). On the other hand, people with systemic lupus erythematosus were significantly more often female and younger (*p* < 0.001, *p* < 0.001, respectively). A case comparison is presented in Table 1.

### 3.2. Comparison of Mental Health and Sleep Factors between the Group of People with Systemic Lupus Erythematosus and People without a Diagnosis of Systemic Lupus Erythematosus

The survey involved 723 respondents. In total, 507 participants (70.1%) suffered from anxiety according to the GAD-7 score (>4 points). In total, 586 participants (81%) showed depressive symptoms on the PHQ-9 scale (>4 points). In total, 514 (71.1%) participants had insomnia according to the score of the ISI scale (>8 points). People with SLE were significantly more often had symptoms of anxiety, depression and sleep disorders compared to those without SLE (*p* < 0.001, *p* < 0.001, *p* < 0.001 respectively). These individuals with SLE also significantly more often demonstrated higher scores on all three scales (GAD-7, PHQ-9, and ISI) compared to those without SLE (*p* < 0.001, *p* < 0.001, *p* < 0.001 respectively). A case comparison is presented in Table 2.

### 3.3. Comparison of Mental Health and Sleep Factors after Adjustment for Covariates

Due to the low frequency of occurrence, the analysis did not take into account coronary heart diseases, heart failure, and chronic obstructive pulmonary disease.

The analysis of the univariable logistic regression model showed that a diagnosis of SLE was associated with a much higher risk of elevated anxiety on the GAD-7 scale (OR = 3.443; *p* < 0.001), depression on the PHQ-9 scale (OR = 4.095; *p* < 0.001), and sleep disorders on the ISI scale (OR = 5.032; *p* < 0.001). After the results were adjusted by age, gender, and diagnosis of the following diseases: hypertension, diabetes mellitus, dyslipidemia, and cigarette smoking, the increased risk of anxiety on the GAD-7 scale (OR = 3.683; *p* < 0.001), depression on the PHQ-9 scale (OR = 4.183; *p* < 0.001), and sleep disorders on the ISI scale (OR = 6.781; *p* < 0.001) were confirmed. The results are presented in Table 3.

## 4. Discussion

When searching medical databases, we did not find any other study that compared mental health factors during the SARS-CoV-2 pandemic between people with and without SLE.

The current outbreak of SARS-CoV-2 infection has led to global changes in many dimensions of our daily lives. Ubiquitous information about the number of deaths, new diseases, diseases predisposing one to a severe and unfavorable course of SARS-CoV-2 infection, lack of targeted treatment, social isolation, change of existing habits, long-term quarantine, and a limited or total lack of access to goods or medical support does not only raise public health concerns but also causes a huge mental burden by generating emotional stress and elevated anxiety. It is widely assumed that the pandemic has aggravated depression, anxiety, and related sleep disorders.

In this study, a significant number of participants experienced symptoms of anxiety, depression, and insomnia, with a 100% prevalence of these symptoms in the group of people with SLE—this is different from the frequency of these disorders in other studies [8,12,27,28]. This may be due to the fact that our study was conducted in Europe where an infection problem on such a scale has not occurred for nearly 100 years, while the aforementioned studies were conducted in Asia where epidemics have been more frequent. This has meant that the institutions responsible for organizing the protection of public safety in China, Hong-Kong, and Taiwan have well-prepared procedures, well-trained service personnel, as well as an entire industry that is capable of adapting to the goal of fighting a pandemic. These differences may also arise from the different tools used to assess depression, anxiety, or sleep disorders and from the differences in the project design itself.

Epidemiological data provide evidence of a steady increase in autoimmune diseases over the last decade. It has long been known that autoimmune diseases show clear gender differences [29], with many more women contracting these diseases than men [30]. The effect of age varies, depending on the disease. According to the study by Fairwather et al., the majority of autoimmune diseases manifest themselves before the age of 50 and are characterized by acute cellular pathology, whereas those manifesting after 50 are characterized by chronic inflammation and fibrosis [31]. These findings concerning age and gender are consistent with the findings of this study, in which people with a diagnosed SLE were significantly younger and more frequently female compared to people without the SLE.

In recent years, many studies have been conducted on the relationship between anxiety, depression, and sleep disorders in patients with chronic diseases. It has been observed that patients with chronic diseases often exhibit mental disorders [32]. In the study conducted by Polukandrioti et al., it was noted that at least 20% of patients with coronary artery disease showed symptoms of anxiety and depression [33]. Other studies confirmed the same relationship in patients with chronic neurological diseases, chronic pain, kidney diseases, and respiratory diseases [34,35,36,37,38,39].

The most important discovery of our study is the fact that despite the significantly more frequent occurrences of chronic diseases in the group of people without a diagnosis of SLE, it was the group of patients with SLE that showed elevated symptoms of anxiety, depression, and sleep disorders (*p* < 0.001, *p* < 0.001, *p* < 0.001, respectively). These individuals, when the results were adjusted for age, gender, chronic diseases, and smoking, showed more than a 3.6 fold increase in the risk of anxiety symptoms, a more than 4.1 fold increase in the severity of depression symptoms, and more than a 6.7 fold increase in the severity of sleep disorders. An autoimmune disease in itself is a source of many stress factors, including reduced activity and fulfillment of social roles at home and work, financial difficulties related to reduced income and high costs of medical care, changes in external appearance resulting, among others, from complications of the applied immunosuppressive treatment, loss of independence, and impaired interpersonal relations [40]. Even slight intensification of the influence of everyday stress factors in people with autoimmune diseases may intensify the symptoms of the disease, i.e., adversely affecting the time of remission [41]. 

This is why patients with a diagnosis of SLE, even after adjusting for those differences in co-occurring conditions have more severity anxiety, depression, and sleep disorders, as confirmed by our study. 

Our study had several limitations. First of all, it lacked longitudinal observations in people included in this study. Secondly, the number of respondents was limited. Thirdly, it was impossible to determine a correlation between the presence of anxiety, depression, insomnia, and a specific type of other autoimmune disease. Fourthly potential selection bias in who participated in the survey. This means that a longitudinal, multi-center study with a greater number of respondents is required.

## 5. Conclusions

In patients with a diagnosed SLE in the era of SARS-CoV-2 there is a higher risk of exacerbation of anxiety, depression, and sleep disorders than in patients with other chronic diseases. It is precisely these people, in a state of mental decompensation, who require informational support, medical support, stress reduction, and rest. Therefore, the mental health of patients with SLE in the times of the SARS-CoV-2 pandemic is not only an extremely important medical problem but also a social one and must require special attention.

## Figures and Tables

**Table 1 jcm-09-02872-t001:** Comparison of selected parameters in patients with and without systemic lupus erythematosus.

		Control (*n* = 589)	SLE (*n* = 134)	*P*
Sex	female	275 (46.69)	119 (88.81)	<0.001
	male	314 (53.31)	15 (11.19)	
Age [years], mean ± SD; Me		39.71 ± 7.07; 39.00	38.34 ± 5.62; 38.00	<0.001
Do you have hypertension? (*n*, %)	No	487 (82.68)	125 (93.28)	0.003
	Yes	102 (17.32)	9 (6.72)	
Do you have diabetes mellitus? (*n*, %)	No	574 (97.45)	133 (99.25)	0.340
	Yes	15 (2.55)	1 (0.75)	
Do you have coronary heart disease? (*n*, %)	No	588 (99.83)	133 (99.25)	0.814
	Yes	1 (0.17)	1 (0.75)	
Are you suffering from heart failure? (*n*, %)	No	587 (99.66)	134 (100.00)	0.814
	Yes	2 (0.34)	0 (0.00)	
Do you have dyslipidemia? (*n*, %)	No	470 (79.80)	115 (85.82)	0.139
	Yes	119 (20.20)	19 (14.18)	
Do you have the chronic obstructive pulmonary disease? (*n*, %)	No	586 (99.49)	134 (100.00)	0.934
	Yes	3 (0.51)	0 (0.00)	
Do you smoke cigarettes? (*n*, %)	No	329 (55.86)	130 (97.01)	<0.001
	Yes	260 (44.14)	4 (2.99)	

Abbreviations: *p*—statistical significance, *n*—number of patients, Me—median, SD—standard deviation.

**Table 2 jcm-09-02872-t002:** Comparison of the severity of anxiety, depression, and sleep disorders in patients with and without systemic lupus erythematosus during the SARS-CoV-2 pandemic.

		Control (*n* = 589)	SLE (*n* = 134)	*P*
GAD-7	Mean ± SD; Me	6.56±4.06; 6.00	19.66±1.10; 20.00	<0.001
	≤4	216 (36.67)	0 (0.00)	<0.001
	>4	373 (63.33)	134 (100.00)	
PHQ-9	Mean ± SD; Me	8.65 ± 4.27; 9.00	20.61 ± 2.01; 21.00	<0.001
	≤ 4	137 (23.26)	0 (0.00)	<0.001
	>4	452 (76.74)	134 (100.00)	
ISI	Mean ± SD; Me	10.18 ± 4.96; 11.00	22.66 ± 2.30; 23.00	<0.001
	≤ 8	209 (35.48)	0 (0.00)	<0.001
	>8	380 (64.52)	134 (100.00)	

Abbreviations: GAD-7—Generalized Anxiety Disorder scale; PHQ-9—Patient Health Questionnaire, ISI—Insomnia Severity Index, *p*—statistical significance, *n*—number of patients, Me—median, SD—standard deviation.

**Table 3 jcm-09-02872-t003:** Logistic regression model of the severity of mental health disorders in patients with and without systemic lupus erythematosus.

	SLE (No Adjusted)			*p*	SLE (Adjusted by Potentially Distorting)			*p*
	OR	Cl −95%	Cl +95%		OR	Cl −95%	Cl +95%	
GAD7	3.443	2.409	4.921	<0.001	3.683	2.271	5.974	<0.001
PHQ-9	4.095	2.786	6.020	<0.001	4.183	2.544	6.878	<0.001
ISI	5.032	3.184	7.954	<0.001	6.781	2.968	15.492	<0.001

Abbreviations: GAD-7—Generalized Anxiety Disorder scale; PHQ-9—Patient Health Questionnaire, ISI—Insomnia Severity Index, *p*—statistical significance, OR—odds ratio, CI—confidence interval. Notes: Potentially distorting data (age, gender, diagnosed hypertension, diabetes mellitus, dyslipidemia and cigarette smoking).

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
