# Peer review of "Evaluation of Mental Health Factors among People with Systemic Lupus Erythematosus during the SARS-CoV-2 Pandemic"

_jcm, 2020, doi:10.3390/jcm9092872_

Round 1
Reviewer 1 Report
Overall, this is an excellent paper examining mental distress and sleep difficulties among patient with SLE during the COVID-19 pandemic. I am not aware of any literature showing similar findings and think the implications of this work are important. The paper is well written and the methods are mostly sound, though I do have some questions and suggestions:
Introduction
-I think the introduction could be strengthened by making more explicit why it is important to look at mental health during the pandemic among people with autoimmune disease, i.e. why these individuals may be more at risk (e.g. greater fear of being infected, more isolated because avoiding more people and public places, etc.)
Materials and Methods
-Was this an online or in person survey? If online, please provide more info about survey platform.
-Were all patients with SLE from those 6 hospitals recruited? If not, what proportion of eligible individuals participated? Do you have any data on differences between those who did and did not participate?
-Please specify the exact cut-offs you used for the GAD-7, PHQ-9, and ISI, for readers who may not be familiar with those, and cite this literature. I see from your results that used cut offs for the GAD-7 and PHQ-9 and cut off of 8 for the ISI. What was the rationale for this? Given the 100% prevalence of any symptoms in the SLE group, perhaps using a different cut off would be more informative?
-Were co-occurring diseases assessed via self-report of diagnosis or confirmed with medical record? Is the wording in Table 1 the exact wording used in the survey? Please clarity so the reader is clear how the covariates and health conditions were defined/assessed.
-Why were those specific covariates used in the regression model? Are they conceptualized as confounders? Please provide more rationale for inclusion of those specific health conditions and tobacco. Are there other covariates, especially those having to do with socioeconomic status, collected?
Results:
-What proportion of the study sample, especially those with SLE, were currently hospitalized, if any?
-Section 3.2 title – I think it would be clearer to call this ‘Mental Health and Sleep Factors” rather than “measurement results”
-The GAD-7, PHQ-9, and ISI provide info about symptoms but not diagnosis or necessarily even presence of disorder. Please adjust the language in the results to reflect the severity of symptoms, for example, ‘moderate depressive symptoms,’ ‘severe symptoms of anxiety’, rather than saying that patients “suffered from anxiety” which makes it seem like they have an anxiety disorder. Similarly, please make sure titles for Tables 3 and 4 and language throughout the paper do not refer to ‘severity of mental health disorders’.
-Section 3.3 title – section 3.2 also has to do with comparison of mental health. Perhaps 3.3 could be renamed to ‘Comparison of mental health and sleep factors after adjustment for covariates” or something to that effect?
-Table 4: Please add which covariates were adjusted for in the legend. Why were demographic factors (age, gender, etc.) not adjusted for?
-I am confused about what the Odds Ratios represent in Tables 3 and 4. My understanding from the description of your paper that the OR should be the odds of GAD-7 anxiety among those with SLE compared to those without SLE, in which case one single OR should be provided for each mental health measure. But an OR is provided for both the control and the SLE groups. Please clarify.
Discussion
-Is there data pre-COVID-19 on GAD-7, PHQ-9, and ISI among patients with SLE, in order to assess whether these are typical numbers for this population? Similarly, comparing control estimates to those of the general population in other studies during the pandemic and also other studies prior to the pandemic would be helpful for context.
-The sentence ‘this is 152 different from the frequency of these disorders in other studies” is a bit too simple – please expand on how your study results differ from those in the studies you cited. Is the control group MH worse in your study relative to these studies? There are also US estimates available, which may be more comparable than those from the Asian countries.
-The discussion paragraph on age and gender doesn’t seem to be super relevant. What would be more compelling is showing how age and gender are related to MH and insomnia in the SLE and control groups.
-The finding of more co-occurring disease is in the control group strikes me as odd – is this in line with the literature? Could this be a selection bias due to people with more severe SLE not responding to the survey?
-“This is why patients with a diagnosis of SLE, constitute the most threatened anxiety, depression 189 and sleep disorders group of patients among all chronic diseases, as confirmed by our study.” --> Unless I missed some data in your paper, I don’t think this conclusion is accurate. You didn’t compare MH and sleep across respondents with different diagnoses, but rather you compared SLE to control and adjusted for co-occurring diseases. I think it is more accurate to say that even after adjusting for those differences in co-occurring conditions, individuals with SLE have more severity MH and insomnia symptoms.
-“ Thirdly, it was impossible to determine a correlation between the presence of anxiety, depression, insomnia, and a specific type of other autoimmune disease.” --> I’m not sure what this means. Do you mean you only looked at SLE and can’t generalize to other autoimmune diseases?
-Other limitation to include: potential selection bias in who participated in the survey?
-“Therefore, the mental health of patients with SLE in the times of the SARS-CoV-2 pandemic is not only an extremely important medical problem but also a social one and must require special attention.”--> can you expand on this a bit? What have you learned from showing that MH distress and insomnia are more severe in this population during the pandemic? Why do you think this is the case? Why does it matter? What are the implications?
Reviewer 2 Report
The COVID 19 pandemy is emerging issue of modern world. Except on the impact of physical health the really important is mental health. Wankowicz et al. checked the impact of COVID-19 on the mental health in patients with Systemic Lupus Erythematosus (SLE). It is very interesting study revealing that the impact of COVID-19 on mental health is bigger than in unaffected subjects. What is more, authors presented also that mental health condition is much worse in COVID19 pandemy than out of it. In previous COVID-19 studies there were some evaluating mental health in SLE and other autoimmune disorders. The study is well developed, the number of subjects is impressive taking into account rarity of the SLE. The conclusions are consisted with the evidence and arguments presented. The language is fine. I have only minor comment for authors to add more detailed paragraph in introduction section about gneral prevalence of mental health conditions in patients with autoimmune disorders (like SLE, MS, for example use following paper to refer Neurol Neurochir Pol Nov-Dec 2018;52(6):704-709)
